# SUV3 Helicase and Mitochondrial Homeostasis

**DOI:** 10.3390/ijms24119233

**Published:** 2023-05-25

**Authors:** Phang-Lang Chen

**Affiliations:** Department of Biological Chemistry, University of California, Irvine, CA 92697, USA; plchen@uci.edu

**Keywords:** mitochondria, degradosome, mtEXO, SUV3, DSS1, PNPase

## Abstract

SUV3 is a nuclear-encoded helicase that is highly conserved and localizes to the mitochondrial matrix. In yeast, loss of SUV3 function leads to the accumulation of group 1 intron transcripts, ultimately resulting in the loss of mitochondrial DNA, causing a petite phenotype. However, the mechanism leading to the loss of mitochondrial DNA remains unknown. SUV3 is essential for survival in higher eukaryotes, and its knockout in mice results in early embryonic lethality. Heterozygous mice exhibit a range of phenotypes, including premature aging and an increased cancer incidence. Furthermore, cells derived from SUV3 heterozygotes or knockdown cultural cells show a reduction in mtDNA. Transient downregulation of SUV3 leads to the formation of R-loops and the accumulation of double-stranded RNA in mitochondria. This review aims to provide an overview of the current knowledge regarding the SUV3-containing complex and discuss its potential mechanism for tumor suppression activity.

## 1. Introduction

Mitochondria are membrane-bound organelles responsible for producing the majority of the cell’s energy supply in the form of adenosine triphosphate (ATP) through oxidative phosphorylation coupled with the Krebs cycle. This process creates a proton gradient across the mitochondrial inner membrane, ultimately producing ATP. Respiration enables mitochondria to convert nutrients into ATP, providing the energy necessary for various cellular processes [1].

Beyond energy production, mitochondria also play a crucial role in regulating cytosolic calcium levels. Calcium is a vital signaling molecule that participates in various cellular processes, such as muscle contraction, nerve signaling, and cell division. Mitochondria help regulate calcium ion levels by taking excess calcium from the cytoplasm and sequestering it into their matrix. Maintaining appropriate calcium levels is crucial for cellular functions, and disruptions can lead to various pathologies, such as muscle weakness, seizures, and neurodegenerative diseases. Therefore, the ability of mitochondria to regulate calcium levels is critical for the proper functioning of cells and tissues [2,3].

In addition, mitochondria also play a crucial role in regulating the production of reactive oxygen species (ROS) [4,5]. ROS are chemically reactive molecules that contain oxygen, such as superoxide anion, hydrogen peroxide, and hydroxyl radicals. These molecules are generated as byproducts of cellular metabolism, particularly during oxidative phosphorylation in the mitochondria. While some ROS molecules are essential for various cellular signaling pathways, high levels of ROS can cause damage to cellular components, such as DNA, proteins, and lipids, leading to oxidative stress and cellular dysfunction. Therefore, the production of ROS must be tightly regulated to maintain cellular homeostasis. Mitochondria have a complex network of enzymes and proteins that are involved in the regulation of ROS production. These include enzymes, such as superoxide dismutase, catalase, and glutathione peroxidase, which neutralize ROS molecules by converting them into less harmful products [5,6].

Another critical function of mitochondria is fatty acid beta-oxidation, which breaks down fatty acids to generate energy [7]. This process occurs in the mitochondria’s matrix. It involves a series of enzymatic reactions that cleave the fatty acid molecules into two-carbon fragments, which are then converted to acetyl-CoA. This molecule enters the Krebs cycle for further energy production. Fatty acid beta-oxidation is a crucial energy source for the heart, skeletal muscle, and kidneys. In addition, fatty acid beta-oxidation provides the energy source during periods of fasting. It also plays a critical role in maintaining cellular homeostasis by regulating lipid metabolism and providing substrates for membrane synthesis and cellular signaling.

Furthermore, mounting evidence indicates that mitochondria play a crucial role in the innate immune response and the defense against viral infections [8,9,10,11,12]. For instance, the mitochondrial antiviral signaling protein (MAVS) can activate the type I interferon (IFN) pathway, which is a crucial mechanism for antiviral defense [12]. Additionally, mitochondria modulate the activity of antiviral signaling pathways and regulate the production of ROS, which can influence the immune response to viral infections [9,10,12] and lead to impaired immune system responses [13,14].

As previously discussed, mitochondria play a critical role in maintaining cellular homeostasis. Therefore, it is unsurprising that mitochondrial dysfunction has been associated with the pathogenesis of various diseases, such as cancer, diabetes, and neurodegenerative disorders [15,16,17,18,19,20]. Further research is necessary to improve our understanding of the mechanisms governing mitochondrial function. Such research could provide valuable insights into developing new therapies and treatments for diseases related to mitochondrial dysfunction.

## 2. Mitochondrial Gene Expression Control

The mitochondrion is a unique organelle with a genome with limited coding capacity and relies on various nuclear-encoded accessory factors for replication, transcription, and translation [21]. Figure 1 illustrates the critical steps in the highly controlled and intricate process of mitochondrial transcription regulation. This process involves numerous nuclear-encoded factors, including transcription factors, RNA processing factors, and RNA surveillance machinery (mtRS). In the following paragraphs, the essential steps will be briefly overviewed. For further insights, refer to a comprehensive review by Shokolenko et al. [22].

Mitochondrial transcription is initiated by the DNA-directed RNA polymerase (POLRMT), which transcribes mitochondrial genes, including those that encode mitochondrial tRNAs and rRNAs [23,24,25]. One crucial protein involved in regulating the initiation of mitochondrial RNA transcription and replication and maintenance of mitochondrial DNA is mitochondrial transcription factor A (TFAM) [26,27]. TFAM interacts with POLRMT to form the preinitiation complex [28]. In addition to TFAM, the mitochondrial transcription factors B1 and B2 (TFB1M and TFB2M) [28,29] bind to POLRMT, resulting in a conformational change that leads to promoter unwinding and initiation of transcription [30]. After initiation, the transcription process requires the mitochondrial transcription elongator factor (TEFM), a single-strand binding protein (SSBP), and the family of mitochondrial transcription termination factors (mTERF1-4) to complete the long polycistronic transcript [31,32].

Mitochondrial RNA is transcribed as a polycistronic long RNA from both Heavy-Strand and Light-Strand, encoding two rRNAs, 22 tRNAs, and 13 proteins that are components of oxidative phosphorylation [21]. The long polycistronic RNA requires mitochondrial RNA processing factors, including cleaving, polyadenylation, editing, and degradation (Figure 1). These factors ensure that mitochondrial RNAs are adequately processed and modified and maintain their integrity and functionality [33]. For instance, the 5′-end of mt-tRNAs is processed by the protein complex RNaseP, composed of MRPP1, −2, and −3 in the granules [34,35]. RNaseZ (also named Zinc phosphodiesterase ELAC protein 2 (ELAC2)) is responsible for the 3′-end processing of the mt-tRNAs [36,37].

The RNA-binding protein LRPPRC (leucine-rich pentatricopeptide repeat-containing protein) is another crucial factor in mitochondrial RNA processing. LRPPRC plays a role in stabilizing and processing several mitochondrial mRNAs and is involved in assembling mitochondrial ribosomes [38]. Along with transcription and processing factors, the mitochondrial RNA surveillance machinery (mtRS) also plays a critical role in ensuring the quality control of mitochondrial RNA molecules (as shown in Figure 2). The mtRS system monitors and degrades improperly processed or damaged mitochondrial RNAs to ensure the proper functioning of the mitochondrial genome [39].

## 3. The Mitochondrial RNA Surveillance System

The mitochondrial RNA surveillance system (mtRS) is a highly conserved mechanism, modeled after the *E. coli* degradosome, which consists of ribonucleases, such as RNaseE and PNPase, an RNA helicase called RhlB, and various accessory proteins [45]. This activity is present in all forms of life (see Table 1). In budding yeast, the degradosome, known as mtEXO, is composed of SUV3, an RNA helicase, and DSS1, a ribonuclease, which work together to aid in RNA maturation and break down damaged or unnecessary mitochondrial RNA molecules [46,47]. In higher eukaryotes, including humans, the mitochondrial RNA degradation system is more complex and involves other critical components, such as the exonuclease PNPase and mtPAP, although the exact mechanism of action remains to be experimentally validated [43,44]. SUV3 remains a vital player in this pathway as an RNA helicase that unwinds RNA secondary structures, enabling access to other RNA-degrading enzymes. In addition to mtEXO, multiple other ribonucleases contribute to the mtRS system to maintain the integrity and function of mitochondrial RNA molecules. For example, human REXO2 degrades short mitochondrial RNA molecules that are produced during the processing of long polycistronic mtRNA [48]. Any dysregulation or malfunctioning of this mtRS system can cause the accumulation of improperly processed or damaged mitochondrial RNAs, leading to mitochondrial dysfunction and disease [49]. Overall, degradosomes play crucial roles in RNA metabolism and quality control across all domains of life, but there are significant differences in their composition and function across different organisms.

### 3.1. Bacterial Degradosome

The bacterial degradosome is a multienzyme complex that participates in RNA processing and degradation in both Gram-positive and Gram-negative bacteria [50,54]. In *E. coli*, the complex comprises various proteins, including RNase E, PNPase, RhlB helicase, and enolase. RNase E, the central component of the degradosome, is an endonuclease that cleaves RNA substrates. PNPase is an exoribonuclease that degrades RNA from the 3′ end and exhibits polyadenylation activity. RhlB helicase is an RNA helicase that unwinds RNA duplexes, facilitating access to RNase E and PNPase for RNA degradation. Although enolase is a glycolytic enzyme that interacts with the degradosome, its exact role in regulating degradosome activity remains unclear. The number and composition of degradosomes vary between bacterial species [54]. The bacterial degradosome is auto-regulated, tightly controlled post-translationally, and compartmentalized [54]. Notably, the bacterial degradosome plays a crucial role in maintaining RNA homeostasis and regulating gene expression in bacteria, as demonstrated by numerous studies [45,50].

### 3.2. Budding Yeast Degradosome

The yeast mitochondrial degradosome is a multi-protein complex responsible for RNA turnover and quality control. The essential degradosome components include RNA helicases and exonucleases, such as SUV3 and DSS1, which work together to ensure proper mitochondrial RNA degradation. SUV3 was initially identified through its ability to suppress the accumulation of the omega intron in SUV3-1 yeast strains [55]. Further analysis of its gene sequence revealed that it encodes a putative ATP-dependent RNA helicase, and subsequent studies have shown that SUV3 is an evolutionarily conserved nuclear-encoded mitochondrial-localized helicase essential for mitochondrial function in eukaryotes. The degradosome’s concerted action, with SUV3 unwinding RNA secondary structures and DSS1 degrading RNA from the 3′ end, is critical for preventing the accumulation of aberrant RNA molecules that could impede mitochondrial function, ensuring proper mitochondrial RNA turnover [47].

### 3.3. Fission Yeast Degradosome

*S. pombe* has a protein called Rpm2/Pah1, similar to the RNA helicase SUV3, and can be exchanged between *S. cerevisiae* and ^S. pombe^ [51]. However, Rpm1/Par1, the counterpart of exoribonuclease DSS1, is specific to Pombe yeast. The primary defect observed in fission yeast knockout strains is their inability to perform downstream processing of transcripts. Furthermore, the absence of Pah1 leads to the instability of mitochondrial RNA ends. Hoffmann et al. suggested that both fission yeast proteins play a minor role in mitochondrial RNA degradation [51]. These observations indicate that other fission yeast mitochondria enzymes may have taken over the RNA-degrading function.

### 3.4. Plant Degradosome

Plants have different RNA surveillance systems within their mitochondria and chloroplasts, consisting of various RNA-binding proteins and ribonucleases involved in RNA turnover and degradation [52]. While bacterial and yeast degradosomes have been extensively studied, the plant mitochondrial degradosome needs to be better understood. One component of the plant mitochondrial degradosome is the SUV3 protein, which is homologous to the yeast mitochondrial RNA helicase SUV3. The gene encoding the SUV3 protein can be found in the genome of Arabidopsis thaliana, a model plant species used in molecular biology research. Additionally, the plant mitochondrial degradosome includes several other proteins, such as the RNase R protein, which is also involved in RNA degradation, and the polynucleotide phosphorylase (PNPase) protein, which has both exoribonuclease and poly (A) polymerase activity. Although the plant mitochondrial degradosome is still not fully understood, the presence of the SUV3 homolog in Arabidopsis thaliana suggests that this protein may play a crucial role in plant mitochondrial RNA turnover and degradation [52].

The chloroplast has proteins that function in RNA turnover and quality control, as the organelle derived from a cyanobacterial ancestor whose RNA surveillance system shares similar traits with prokaryotes [53]. Similar to the prokaryote, the major chloroplast endoribonucleases found are identical to RNase E, namely, RNase J. Researchers have suggested that RNase J has assumed the surveillance role in chloroplasts. At the same time, due to the lack of RNase J, mitochondria depend on PNPase for the same function. The chloroplast RNA quality control system also includes RNase R, glycolytic enzyme enolase, and a putative RNA-binding protein. While the plant SUV3 protein has not been detected in the chloroplast, another helicase function may be present to unwind structured RNA and assist the action of exoribonucleases.

Growing evidence suggests that chloroplast RNA stability plays a significant role in plant responses to environmental stresses [56,57]. However, how the RNA surveillance machinery responds to and detects RNA under various ecological conditions remains to be investigated. Despite this, multiple ribonuclease-containing complexes in the chloroplast are known to be critical for degrading aberrant or unnecessary chloroplast RNAs, processing chloroplast transcripts, and maturing ribosomal RNA, as well as regulating plastid gene expression. These functions are essential for maintaining proper chloroplast function and ensuring plant survival in changing environmental conditions.

### 3.5. Mammalian Degradosome

The human SUV3 protein, encoded by the *SUPV3L1* gene, is a Suv3-like RNA helicase located on chromosome 10. There is significant sequence homology between the yeast Suv3 gene and mammalian SUV3 gene, as depicted in Figure 2 [40,41,42]. While the complete identification of the mitochondrial degradosome complex in higher eukaryotes is still ongoing, several experimental studies have indicated that the SUV3 helicase plays a regulatory role in mitochondrial RNA degradation [58,59,60,61,62]. In a reconstituted system, SUV3 has been observed to form a complex with PNPase to facilitate RNA degradation [43,44]. It is important to note that these two proteins localize in different subcellular compartments, with SUV3 primarily found in the mitochondrial matrix and PNPase in the inner membrane space [63].

Borowski et al. have shown that a small portion of PNPase is present in the mitochondrial matrix. It interacts with SUV3 to form the mtEXO complex, also known as the mitochondrial degradosome [60]. The experimental evidence has suggested that SUV3 and PNPase share distinct functions in mitochondrial RNA degradation. While both proteins are necessary for efficient RNA turnover, SUV3 is more crucial in breaking down structured RNAs containing stem-loop structures. In contrast, PNPase is primarily involved in degrading single-stranded RNA.

Although the exact composition of the mitochondrial degradosome complex is still unknown in higher eukaryotes, studies have demonstrated that SUV3 plays a crucial role in regulating mitochondrial RNA degradation. Specifically, SUV3 is an RNA helicase that unwinds RNA secondary structures. It is a critical component of the mtEXO complex, formed by the association of SUV3 with a small fraction of the mitochondrial RNA degrader PNPase.

The mtEXO complex is essential for efficient RNA turnover and the maintenance of mitochondrial function. It degrades aberrant or unneeded mitochondrial RNAs and processes primary transcripts to generate mature mRNAs and rRNAs. This process is critical in highly eukaryotic systems, where circular mtDNA is transcribed from both directions, creating a perfect chance to form dsRNA if the process is not regulated. Defects in the mtEXO complex can lead to mitochondrial dysfunction, which can cause various diseases, including neurodegenerative disorders.

Further investigation is necessary to fully understand the intricate mechanisms involved in mitochondrial RNA degradation and the role of SUV3 and PNPase in this process. Researchers are actively exploring the function of other components of the mitochondrial degradosome complex and their interactions with SUV3 and PNPase. Additionally, regulating the complex in response to environmental stresses and developmental changes remains an active area of research.

## 4. Dysfunction of Degradosome: Recent Developments

The dysfunction of degradosomes can lead to different consequences depending on their specific function. When degradosomes fail, RNA fragments and degradation intermediates can accumulate, causing an imbalance in RNA metabolism that can harm cellular processes. In particular, an impaired mitochondrial function can occur, as degradosomes play a critical role in regulating the turnover of mitochondrial RNA and protein. At the organism level, degradosome dysfunction has been linked to various diseases, including cancer, neurodegenerative, and metabolic disorders [49,62,64,65].

Haute et al. provided an excellent review of various factors that influence mitochondrial RNA maturation and their associated disorders, focusing on the crucial role of quality control and RNA stability [49]. Among them, studies suggest that the inactivation of components of the human mitochondrial degradosome may be linked to different mitochondrial disorders. For example, PNPase, encoded by the PNPT1 gene, plays a crucial role in mitochondrial RNA turnover and potentially RNA import into mitochondria [66]. Many studies have identified pathogenic mutations in PNPT1 that cause mitochondrial disorders [65,67,68]. In one study, researchers identified a PNPase mutation in a family with multiple cases of Leigh syndrome.

The mutations p.Arg136His and p.Pro140Leu have the potential to disrupt the active site of the PNPase enzyme, resulting in impaired functionality. This disruption can lead to the accumulation of unprocessed RNA within the mitochondria and ultimately result in impaired mitochondrial function. This resulted in the symptoms of Leigh syndrome, including developmental delays, movement disorders, and respiratory problems [65]. In other reported cases, it has been observed that the homozygous PNPT1 missense mutation (p.Glu475Gly) affects the homo-trimerization of PNPase, leading to impaired RNA import into the mitochondria [68]. However, the precise role of PNPase in human mitochondria remains unclear, and it is unknown if these mutations affect PNPase’s role in mtRNA turnover in the mitochondrial matrix. Nonetheless, these findings indicate that mitochondrial degradosome dysregulation or malfunction can lead to mitochondrial disorders and emphasize the importance of this complex in maintaining mitochondrial function.

It is well-established that the mitochondrial degradosome plays a crucial role in mtRNA surveillance (as depicted in Figure 3). In addition to this function, SUV3 has been shown to regulate mitochondrial gene expression and maintain the integrity of mitochondrial DNA. SUV3 heterozygotes are prone to cancer and exhibit premature aging phenotypes [62]. Recent studies utilizing specific reagents, such as monoclonal antibody S9.6 to detect R-loop [69] and monoclonal antibody J2 to detect double-strand RNA (dsRNA) [69], as well as RNA sequencing to identify accumulated RNA species in response to degradosome downregulation, have provided potential mechanisms explaining how SUV3 dysfunction can lead to these phenotypes.

### 4.1. R-Loop

Detecting and characterizing mitochondrial R-loops has been challenging due to their unstable nature. However, recent studies have shown that ablation of RNase H can lead to the accumulation of 7S RNA and its association with the mitochondrial control region (MCR), resulting in the formation of R-loops that can be easily detected [70,71]. Additionally, R-loops containing LC-RNA (transcribed from the light strand at MCR) can be readily seen with cross-linking. Currently, there is active research to investigate the physiological role of the abundant R-loops containing LC-RNA in the mitochondria. Several studies have shown that dysfunction of the mitochondrial degradosome complex, consisting of SUV3 and PNPase, can lead to the formation and accumulation of R-loops across the mitochondrial genome, potentially contributing to the development of mitochondrial diseases [72,73]. Transient knockdown of SUV3 in HeLa cells resulted in the formation of R-loops in mitochondrial DNA, reducing mitochondrial DNA copy number and compromising mitochondrial respiration [72]. The study also showed that overexpression of SUV3 could reverse these effects.

Similarly, Pietras et al. found that the knockdown of PNPase in human cells resulted in the accumulation of R-loops and RNA fragments, which altered mitochondrial function and could potentially cause cell death [73]. The study also showed that overexpression of PNPase could rescue the effects of PNPase knockdown. Furthermore, Pajak et al. demonstrated that drosophila mtPAP, an enzyme involved in the polyadenylation of mitochondrial RNAs, plays a role in resolving R-loops in mitochondria [74]. The interactions of mtPAP with the mitochondrial degradosome complex are regulated by physiological conditions [43]. Overall, these findings highlight the critical role of the mitochondrial degradosome complex in maintaining mitochondrial genome stability and proper mitochondrial function. Dysregulation or malfunction of this complex can lead to the accumulation of R-loops in the mitochondrial genome, causing DNA damage, genomic instability, and compromised mitochondrial function, which could contribute to the development of mitochondrial disorders and neurodegenerative diseases.

### 4.2. dsRNA

The monoclonal antibody J2 has emerged as a critical tool in quantitating and visualizing dsRNA in different physiological conditions and genetic perturbations [69]. J2 has been used in several studies highlighting the essential role of SUV3 in maintaining mitochondrial RNA homeostasis by preventing the accumulation of dsRNA in the mitochondrial genome of human cells [75]. In particular, the knockdown of SUV3 leads to the formation of R-loops and dsRNA accumulation in mitochondria. The accumulation of dsRNA in mitochondria can trigger the cGAS pathway and activate innate immune responses such as the production of type I interferons, serving as a crucial mechanism for detecting viral infections and activating the innate immune response. Interestingly, Dhir et al. found that SUV3 knockdown did not lead to the release of dsRNA into the cytosol or the activation of the cGAS pathway [76].

In contrast, PNPase knockdown in mouse embryonic fibroblasts resulted in the accumulation of mitochondrial dsRNA released into the cytosol, triggering the cGAS pathway and interferon response. These observations suggest that PNPase uniquely transports RNA molecules across the mitochondrial membranes and into the mitochondria to prevent their release into the cytosol. PNPase, an RNA exonuclease, functions to degrade or modify dsRNA in the mitochondria before it can be released into the cytosol. In contrast, SUV3, an RNA helicase, may play a role in resolving R-loops and unwinding RNA secondary structures within the mitochondria but not necessarily in dsRNA degradation or modification.

The activation of the cGAS pathway due to dsRNA accumulation in the mitochondria highlights the intricate interplay between the mitochondrial and cytosolic nucleic acid sensing pathways and their implications for immune function and disease. Notably, the mechanism by which dsRNA is transported from the mitochondria to the cytosol for cGAS activation needs to be better understood. Further research is required to fully elucidate the role of PNPase and SUV3 in dsRNA processing and transport and their contribution to the cGAS pathway activation. Nonetheless, these findings have significant implications for our understanding of mitochondrial RNA metabolism, innate immunity, and the development of strategies for treating immune-mediated diseases.

### 4.3. 7S RNA, a Candidate In Vivo Substrate for Mtexo

As mentioned earlier, mtRNA accumulation has been observed in response to the knockdown of components of mtExo. Zhu et al. made an exciting discovery linking mtExo to regulating mitochondrial transcription initiation [77]. The authors identified non-coding 7S RNA as a regulator of mitochondrial transcription in mammalian cells through a negative feedback loop. To further investigate the role of mtExo in regulating mitochondrial RNA metabolism and degradation, the authors screened 75 genes linked to these processes using siRNA knockdown. They found that the depletion of SUV3, PNPase, REXO2, and ATAD3B led to an increase in 7S RNA levels. Since PNPase and SUV3 are suggested components of mtExo in mammalian cells, Zhu et al. demonstrated that recombinant mtExo could degrade 7S RNA in vitro, indicating that it may be a target of mtExo in vivo [77]. These recent findings suggest that in addition to its known role in regulating mtRNA degradation, mtExo may also play a crucial role in regulating mitochondrial transcription, as demonstrated by identifying non-coding 7S RNA as a regulator of mitochondrial transcription initiation in mammalian cells through a negative feedback loop.

These studies have shown that dysfunction of the mitochondrial degradosome complex consisting of SUV3, PNPase, and mtPAP can lead to the accumulation of R-loops and dsRNA in the mitochondrial genome and can have detrimental effects on mitochondrial function and genome stability (Figure 2).

## 5. Perspective

Our early work showed high tumor incidence in SUV3 heterozygous mice [62], and mounting evidence suggests that dysregulation of mitochondrial RNA metabolism, including defects in the mtEXO complex, can contribute to tumorigenesis. As mitochondria are essential organelles responsible for cell energy production, dysfunctional mitochondria are proposed to link to various diseases, including cancer, neurodegenerative disorders, metabolic diseases, and aging. In the past, the favored hypothesis suggested that mitochondrial dysfunction can lead to increased reactive oxygen species (ROS) production, triggering inflammatory responses or promoting apoptosis. Another potential link between dysfunctional mtEXO and inflammation-related tumorigenesis is the activation of the innate immune response. Recent studies suggest that mitochondrial dysfunction can lead to the release of double-stranded RNA (dsRNA) from the mitochondria. The released dsRNA can activate the innate immune system and trigger an inflammatory response. Kim et al. have suggested that the dsRNA from mitochondria can be further released into the extracellular space through exocytosis, which may contribute to inflammation and cancer development [78].

In summary, dysregulation of mitochondrial RNA metabolism, including defects in the mtEXO complex, can lead to mitochondrial dysfunction and proinflammatory signaling pathways that promote tumorigenesis. While the precise mechanisms linking SUV3 knockdown to inflammation-related tumorigenesis require further investigation, it is clear that the mtEXO complex plays a crucial role in maintaining mtRNA homeostasis by degrading aberrant or unwanted mtRNAs in mammalian cells. However, the mechanisms by which mtEXO recognizes and degrades specific mtRNAs and the factors that regulate mtEXO activity still need to be fully understood. Future research may involve identifying additional regulatory factors and binding partners of mtEXO, characterizing the specific RNA substrates targeted by mtEXO, and investigating the role of mtEXO in regulating mtRNA turnover under various cellular stresses. Ultimately, a better understanding of the precise mechanisms underlying mtEXO dysfunction in disease states may provide new opportunities for therapeutic intervention in mitochondrial disorders.

## Figures and Tables

**Figure 1 ijms-24-09233-f001:**
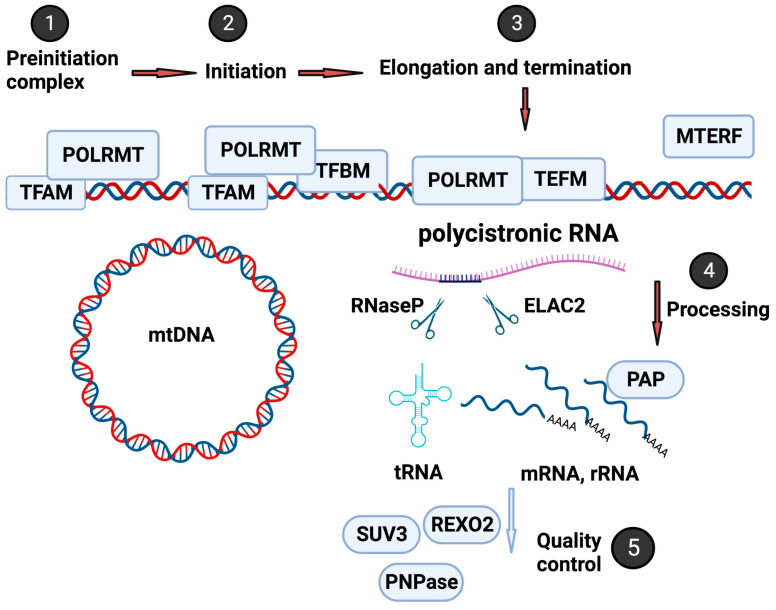
Mitochondrial gene expression control. Mitochondrial gene expression involves transcription by POLRMT, regulated by TFAM, TFB1M, and TFB2M to form a preinitiation complex (Steps 1 and 2), followed by elongation and termination with TEFM, SSBP, and mTERF1-4 (Step 3). The resulting polycistronic mRNA requires processing by factors, including RNaseP and ELAC2, to generate tRNA mRNA with polyadenylation, splicing, and editing (Step 4). The mitochondrial RNA surveillance machinery performs quality control to ensure the proper function of the mitochondrial genome (Step 5). Figure created with biorender.com.“URL (accessed on 19 May 2023)”

**Figure 2 ijms-24-09233-f002:**
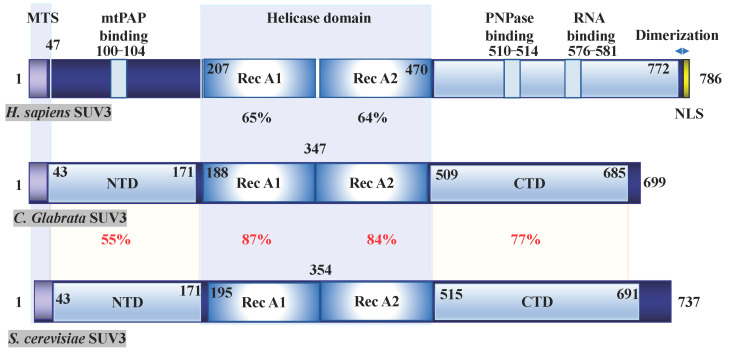
Functional Domain of SUV3. SUV3, found in Homo sapiens (Q8IYB8), Candida glabrata (Q6FKD7), and Saccharomyces cerevisiae (P32580), exhibits domain structure and sequence homology. The protein contains an N-terminal mitochondrial targeting sequence that undergoes cleavage upon entry into the mitochondria. The central core functional helicase domain is highly conserved across all species and can be further divided into RecA1 and RecA2 domains, as elucidated by crystal structural analysis [40,41,42]. Among these regions, the N-terminal domain (NTD) demonstrates the most significant diversity. Research has shown that the deletion of amino acids 100–104 in human SUV3 abolishes its interaction with mtPAP [43]. The C-terminal domain (CTD) is also conserved and has been demonstrated, in the case of yeast SUV3, to coordinate with the RecA1 domain, contributing to DSS1 binding [40]. Additionally, the CTD of human SUV3 has been associated with both PNPase and RNA binding activities [44]. It is worth noting that the C-terminal tail (amino acid 722–786) of the human SUV3 protein is required for dimerization [42]. Furthermore, a putative nuclear localization signal (NLS) has been identified in the human SUV3 protein sequence.

**Figure 3 ijms-24-09233-f003:**
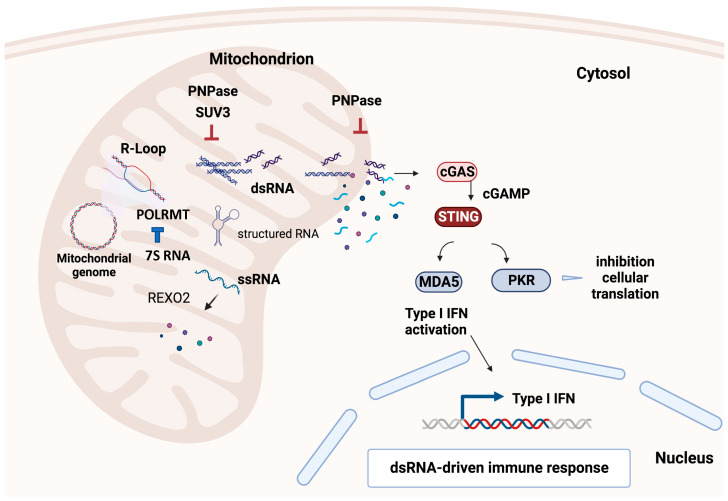
Effects of a faulty degradosome mechanism. When the mitochondrial degradosome complex, made up of SUV3, PNPase, and mtPAP, is knocked down, it can cause a buildup of R-loops, double-stranded RNA (dsRNA), and single-stranded RNA (ssRNA) in the mitochondria. This accumulation can be harmful to mitochondrial function and genome stability. R-loops formed in the mitochondrial genome can promote DNA strand breaks, leading to an unstable mitochondrial genome. dsRNA can leak into the cytosol and activate the inflammatory response and PKR, inhibiting translation. Furthermore, the accumulation of non-coding 7S RNA can inhibit mitochondrial transcription by directly associating with POLRMT. Figure created with biorender.com (accessed on 19 May 2023).

**Table 1 ijms-24-09233-t001:** Components of RNA degradosomes from different species.

Species (Organelle/Genome Size)	Helicase	Exoribonuclease Scaffold	Exoribonuclease	Accessory Proteins	Reference
*E. coli*	Rhlb	RNase E	PNPase	Enolase	[45,50]
*S. cerevisiae* (Mitochondria/85Kb)	SUV3		DSS1		[47]
*S. Pombe* (Mitochondria/19.4Kb)	Rpm2		Rpm1		[51]
*C. glabrata* (Mitochondria/20Kb)	SUV3		DSS1		[40]
*A. thaliana* (Mitochondria/ 366.9Kb)	AtSUV3		PNPase (?)		[52]
*A. thaliana* (Chloroplast/154.5Kb)	?	RNase E RNase J		Enolase	[52,53]
*H. sapiens* (Mitochondria/16.5Kb)	SUV3		PNPase		[44]

? The sequence analysis indicates the presence of SUV3 and PNPase within the A. thaliana genome, suggesting their potential roles as counterparts of plant mitochondrial degradosomes. However, it is important to emphasize that further investigation is necessary to determine the composition and specific functions of these plant mitochondrial degradosomes.

## Data Availability

Not applicable.

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
