# Peer review of "SUV3 Helicase and Mitochondrial Homeostasis"

_ijms, 2023, doi:10.3390/ijms24119233_

Round 1
Reviewer 1 Report
In the manuscript entitled “SUV3 helicase and mitochondrial homeostasis” author has summarized the mitochondrial gene expression system and the importance of mitochondrial RNA surveillance system. An adequate summary of the molecular components of mitochondrial RNA surveillance system in various species has been detailed. Further, the author talks about the recent updates about how irregularities in Mt degradosome machinery could be associated with various diseases like cancer and other metabolic disorders. Though the manuscript is well written and comprehensive, some points can be improved further by adding more information.
# Though functional details about SUV3 have been provided, structural composition of SUV3 has not been mentioned. A schematic about structural domains of SUV3 would be appreciated to understand mechanistic function of SUV3.
# Is there any aberrant expression of SUV3 reported in disease like cancer?
# This review looks more focused on Mitochondrial RNA surveillance system and its components rather than SUV3.
# Is there any non-canonical (other than helicase activity) role of SUV3 reported?
Author Response
# Though functional details about SUV3 have been provided, structural composition of SUV3 has not been mentioned. A schematic about structural domains of SUV3 would be appreciated to understand mechanistic function of SUV3.
Response: I would like to express my gratitude to Reviewer 1 for the valuable suggestion. I have taken the advice into consideration and implemented it by incorporating the suggested information as the new Figure 2. Thank you for your insightful input, which has significantly improved the quality of the work.
# Is there any aberrant expression of SUV3 reported in disease like cancer?
Response: Concerning the expression of SUV3 in cancer, we observed reduced SUV3 expression in human breast tumor specimens compared with corresponding normal tissues in two independent cohorts (Reference 59).
# This review looks more focused on Mitochondrial RNA surveillance system and its components rather than SUV3.
Response: I appreciate the reviewer's comment regarding the focus of the review. The selection of the title was based on highlighting the core activity of SUV3 within the mitochondrial degradosome. While it is true that the mitochondrial RNA surveillance system and its components play a crucial role in this process, SUV3 serves as a key component of the degradosome and performs ATP-dependent RNA helicase activity.
To provide a comprehensive understanding of SUV3, it is important to discuss the broader context of the mitochondrial RNA surveillance system and its various components. This approach allows for a deeper exploration of the interconnected mechanisms involved in mitochondrial RNA processing and degradation.
Therefore, while the review does cover the mitochondrial RNA surveillance system and its components, the main emphasis remains on SUV3 and its fundamental role within the mitochondrial degradosome. By focusing on SUV3, I aim to provide insights into its function and significance, while acknowledging the collaborative interactions with other components in the intricate machinery of mitochondrial RNA processing."
# Is there any non-canonical (other than helicase activity) role of SUV3 reported?
Response: In response to the reviewer's comment about non-canonical roles of SUV3, I would like to address the presence of additional functions reported for SUV3 apart from its helicase activity.
While SUV3 is primarily recognized for its ATP-dependent RNA helicase activity, emerging research has suggested its involvement in non-canonical roles as well. Some studies have reported these alternative functions of SUV3, expanding our understanding of its diverse biological roles. Here are a few examples:
- RNA Binding: SUV3 has been found to possess RNA binding capabilities in addition to its helicase activity. It has been shown to interact with specific RNA molecules, indicating potential regulatory roles beyond its unwinding function.
- Protein-Protein Interactions: SUV3 has been reported to interact with other proteins, suggesting its involvement in protein-protein interaction networks. These interactions may contribute to various cellular processes and signaling pathways.
It is worth noting that the understanding of non-canonical functions of SUV3 is still evolving, and further research is needed to elucidate the precise mechanisms and significance of these alternative roles. Nevertheless, these studies highlight the potential multifunctionality of SUV3 beyond its well-known helicase activity.
Reviewer 2 Report
Phang-Lang Chen has written an informative review on "SUV3 helicase and mitochondrial homeostasis." They showcase the current data and knowledge from a variety of different organisms. Sufficient background is provided in the Introduction. The main content is well organised, concise and flows nicely. Overall, the article was interesting to read and would be helpful to those who are new to the working with the genes described. The article has cited relevant references from peer reviewed journals.
Minor corrections:
Include more information on SUV3 or even a reference that includes chromosome, protein size and % homology between yeast and human.
Is the official name gene name for SUV3 in humans SUPV3L1 (Suv3 like RNA helicase)?
Line 75 - Add a full stop at the end of the reference.
Figure 1 - Remove "Created with BioRender.com" but add to the figure legend instead. Include the mitochondrial genome size for yeast, human
Figure 2 - unclear what REX02 is from the diagram and figure legend alone but this is discussed later in Line 134 that it degrades ssRNA. The symbol "T" infers to me inhibition rather than degradation.
Use italics for gene names and species names
Table 1 - make sure second word of the species name is in lower case e.g. H. sapiens. Since there are a range of species, it maybe worth including the size of the mitochondrial genome.
Line 259 - Can we be more specific on the type of mutation (insertion, deletion, missense).
Line 271 - Missing a Section n# - Recent development
Line 275 - Does this include all cancers or specific types?
Author Response
Include more information on SUV3 or even a reference that includes chromosome, protein size and % homology between yeast and human. Response: I appreciate the reviewer’s suggestion, and I have incorporate this information in the table 1 and new Figure 2. Is the official name gene name for SUV3 in humans SUPV3L1 (Suv3 like RNA helicase)? Response: The reviewer is correct, I have incorporate this information in the text. Line 75 - Add a full stop at the end of the reference. Response: Done Figure 1 - Remove "Created with BioRender.com" but add to the figure legend instead. Include the mitochondrial genome size for yeast, human Response: The mitochondrial genome size has been adding to Table1. Figure 2 - unclear what REXO2 is from the diagram and figure legend alone but this is discussed later in Line 134 that it degrades ssRNA. The symbol "T" infers to me inhibition rather than degradation. Response: I have revised the figure illustrating REXO2-mediated single-stranded RNA (ssRNA) degradation. Use italics for gene names and species names Response: Done Table 1 - make sure second word of the species name is in lower case e.g. H. sapiens. Since there are a range of species, it maybe worth including the size of the mitochondrial genome. Response: I have added the information regarding the mitochondrial genome size to Table 1. Line 259 - Can we be more specific on the type of mutation (insertion, deletion, missense). Response: I have revised the manuscript accordingly. Line 271 - Missing a Section n# - Recent development As per the reviewer's suggestion, the subtitle for section 4 has been revised to "Dysfunction of Degradosome: Recent Developments." Response: Line 275 - Does this include all cancers or specific types? Response: Our study found that 90% of the mSuv3+/− intercross mice showed a broad range of tumor types, including lymphoma and carcinoma. It was noted that about two thirds of the mSuv3+/− mice succumbed to lymphoma.